# Ice Elevation Change Based on GNSS Measurements along the Korth-Traverse in Southern Greenland

Thomas Hitziger [1], Luisa Näke [2] and Karel Pavelka [3,*]

1 Department of Mechanics and Numerical Methods, Institute of Civil and Structural Engineering, Brandenburg University of Technology Cottbus-Senftenberg, 03046 Cottbus, Germany
2 Department of Civil and Mechanical Engineering, Technical University of Denmark, 2800 Kongens Lyngby, Denmark
3 Department of Geomatics, Faculty of Civil Engineering, Czech Technical University in Prague, 16000 Prague, Czech Republic
* Correspondence: pavelka@fsv.cvut.cz

**Abstract:** In 1912, a Swiss expedition led by meteorologist Alfred de Quervain crossed the Greenland ice sheet on a route from Disko Bay to Tasiilaq. Based on that, in 2002, a series of geodetic expeditions carried out by W. Korth and later by T. Hitziger began along the same traverse as in 1912, with the last measurements taken in May 2021. The statically collected GPS/GNSS data provide very accurate elevation changes at 36 points along the almost 700 km long crossing over a period of 19 years. According to this, there is a maximum increase of 2.1 m in the central area and a decrease of up to 38.7 m towards the coasts (influence Ilulissat Isbræ). By using kinematic GNSS measurements, there is a very dense profile with a spacing of a few meters. The comparison of those measurements is performed using crossing points or minimum distances and gives equivalent results for both methods. It is shown that local ice topography is preserved, and thus gaps in data sets can be caught. Areas of accumulation and ablation on the ice sheet can be identified, showing the widespread influence of outlet glaciers up to 200 km. The data can be used for direct verification of altimetry data, such as IceSat. Both IceSat elevations and their changes can be compared.

**Keywords:** Greenland ice sheet; monitoring; GNSS; expedition; Jakobshavn Isbræ; Helheim Glacier; IceSat; climate change; glacier profile





## 1. Introduction

This article is dedicated to the geodesist, polar explorer, and friend Wilfried Korth. He was the project initiator and scientific leader for a long time. The article is also based on his results, so he is mentioned here in memory as an additional author (Figure 1).

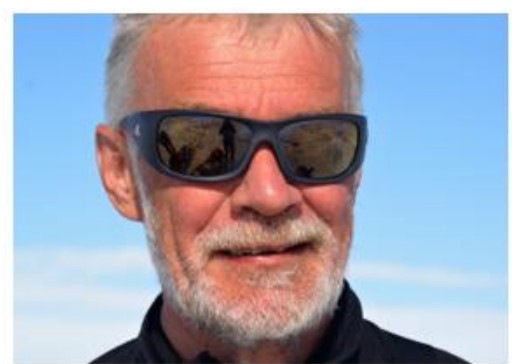

**Figure 1.** Project initiator Wilfried Korth (1959–2019) on his last Greenland Expedition in 2017.

The processes and consequences of climate change have been discussed for many years. Leaving aside catastrophic changes such as volcanic eruptions or earthquakes,

it is indisputable that never in the recent history of the Earth, i.e., in the last million years, have there been fundamentally very rapid changes in the living conditions on our planet [1]. However, the climate today is changing rapidly. Geodetic measurements can make important and precise contributions to the monitoring of changes. In addition to remote sensing technique, which uses a wide variety of technologies, there are also ground-based measurements. These serve as "ground truth" for remote sensing, but because of their accuracy, they can also be used independently.

In the 1970s, only ground-based measurements, often obtained during scientific expeditions or from measuring stations, could be used to monitor the Greenland ice sheet. Aerial methods were also used, but only for coastal parts of Greenland. Here it is worth recalling the pioneering expeditions that began exploring inland Greenland more than a century ago.

One of the first was certainly the expedition of Fridtjof Nansen (1888–1889), who was the first to cross the southern part of Greenland on skis [2]. He brought back a wealth of scientific information and meteorological measurements and proved that the entire Greenland interior was covered by an ice sheet. Another important expedition was undertaken by the Swiss Alfred de Quervain in 1912. De Quervain, a meteorologist, crossed Greenland with three other expedition members considerably further north than Nansen [3]. He was shortly followed by Alfred Wegener and Lauge Koch [4,5].

After the First World War, expeditions were more frequent and much better prepared technically. Airplanes also began to be used for research after the First World War. The systematic mapping of the coastline by the Danish Geodetic Survey in 1931–1934 was significant. The mapping work was carried out using photogrammetry from an aircraft. Today there are thousands of unique photographs in the Danish Airbase project database, which serve as a source of information on the historical state of glaciation [6,7]. Several US military airfields were built in Greenland during World War II, some of which were converted to civilian airfields after the war and are still in use today [8]. Germany also built a small meteorological base in Greenland, but it was destroyed by an American air raid [9]. After the war, economic development began in Greenland, but other US military bases were also built in Greenland during the Cold War. More intensive research on the Greenland ice sheet took place after the fall of the Iron Curtain in the 1990s.

Modern instruments, expedition equipment, and technical support were available, as well as the possibility of using satellite data. The significant progression of global warming and the rapid melting of the western and southern parts of the Greenland ice sheet, in particular, increased interest in research activities [10–12]. Combined data sources and non-traditional technologies like drones, for example, were used in research. Today, drones are the most popular, which allow very detailed measurements in smaller areas, e.g., tracking the movement of a glacier face or capturing the surface with cm resolution [13–15]. Special remote sensing satellites have been used for a long time, since the 1970s, but it's only relatively recently that some data has been free of charge and freely downloadable.

Geodetic satellites monitor gravity changes, radar satellites can use InSAR technology to determine displacements or create digital surface models, and optical satellite systems can help monitor the extent of glaciation [13,16]. Fast and accurate GNSS instruments can monitor the height or movement of glaciers [17–19]. In the context of Arctic polar research, it is worth remembering Nansen's unique polar expedition on the Fram ship (1893–1896); this was followed in 2019 by an international expedition aboard the modern research ship Polarstern. The aim was the comprehensive mapping of the Arctic and, in particular, research on global warming [20]. Glacier changes related to global warming have been investigated in many other scientific papers [21,22].

Twenty years ago, in the summer of 2002, geodesist Wilfried Korth (Figure 1) started a climate research project in Greenland. The main objective was to determine elevations and their changes along a profile across the Greenland ice sheet. A 700 km traverse was surveyed between Tasiilaq on the east coast and Ilulissat on the west coast of Greenland (see Figure 2). After his tragic death, however, some members of his expeditions continued his

work. This provided another valuable amount of information on the changing Greenland ice sheet. The results from all the expeditions are summarized in the following text.

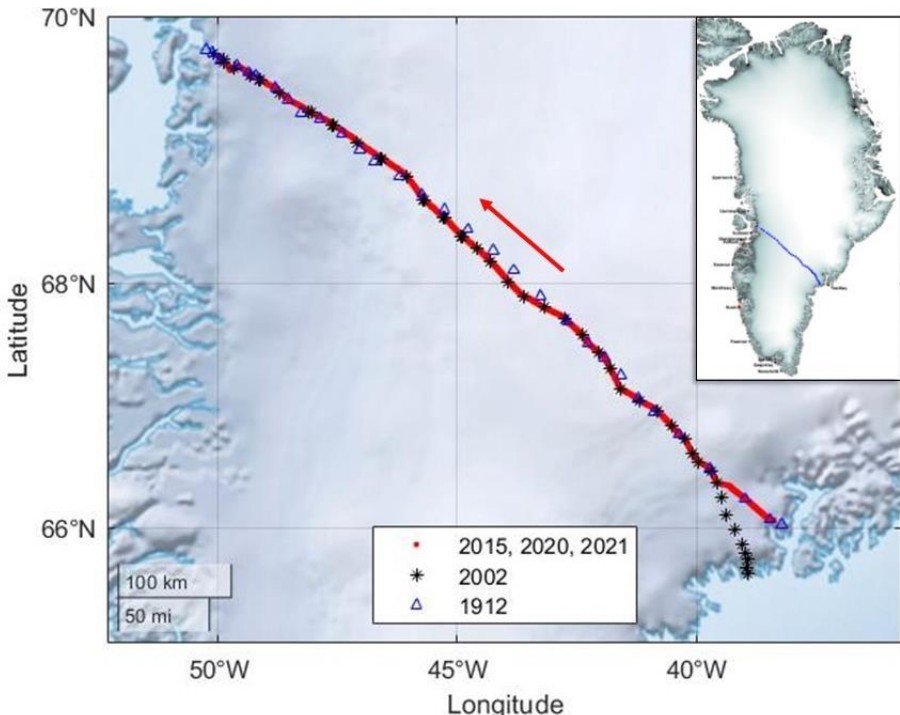

**Figure 2.** Map of Greenland [17] and route of Greenland Korth Expedition (GKE) with walking direction, camps from 2002 and historical camps from 1912. The blue line approximately marks the catchment area of the Helheim and Ilulissat (Jakobshavn) glaciers and the top of the Greenland ice sheet.

This route was first successfully crossed by the Swiss Alfred de Quervain in 1912. Even if the accuracy of his measurements was only relatively low compared to today's possibilities, the large time difference of more than a hundred years naturally tempts a comparison [18], which is especially interesting in the strongly changing marginal area of the ice sheet.

Meanwhile, during the eight expeditions since 2002, the profile was surveyed five times completely and three times partially with high accuracy (see Table 1). The process results in surface elevations with a measurement accuracy of 3–5 cm, from which annual surface changes are derived with similar accuracy. The measurements were carried out between the end of July and the beginning of September. During this period, the summer thaw was ending, while the winter snowfall had not yet begun. It is, therefore, the time of the year when the seasonal variations in ice elevations reach their minimum.

**Table 1.** Overview of geoscientific Expeditions on the historic route.

| Year | Scientific Director | Method of Measurement | Remark |
|------|---------------------|-----------------------|--------|
| 1912 | A. de Quervain | barometric | 39 camps; accuracy in the coastal area +/−3–5 m |
| 2002 | W. Korth | GPS static | 34 positions; +/−3 cm |
| 2006 | W. Korth | GPS static | 34 positions; +/−3 cm |
| 2010 | W. Korth | GNSS static | 34 positions; +/−2 cm |
| 2012 | W. Korth | GNSS static | only east coast; 17 positions +/−2 cm |
| 2015 | W. Korth | GNSS kinematic and static | continuous profile; spot spacing 2–6 m; 700 km; +/−3 cm |
| 2017 | W. Korth | GNSS kinematic | continuous pr.; spot spacing 2–6 m; approx. 180 km; +/−3 cm |
| 2020 | T. Hitziger | GNSS kinematic | continuous pr.; spot spacing 2–6 m; approx. 500 km; +/−3 cm |
| 2021 | T. Hitziger/J. Heim | GNSS kinematic | continuous pr.; spot spacing 2–6 m; approx. 680 km; +/−3 cm |

Based on GNSS technology progress since the 2015 expedition, the measurement program was changed. Unlike in previous years, not only were the profile points de-

termined but the measurements were carried out continuously along the entire route at 1-second intervals. Thus, for the first time, a 700 km long profile with a point spacing of less than 2 m is available. The possibilities for comparison with satellite data have thus improved enormously.

Measurements on the ice sheet can be carried out in very different ways with today's technical possibilities. However, extreme problems occur, especially in the marginal areas, which lead to limitations: the use of (heavy) snowmobiles is hardly possible because of the numerous crevasses, which are often blown, and impossible in the large areas with melt-water rivers, gullies, and ice humps. But this concerns the most interesting area, about 20–30% of the planned route.

As a logistical alternative, skis and pulkas (freight sledges) were used on all expeditions, and the routes were covered on foot. What appears at first glance to be an increased risk is, on closer inspection, a gain in safety. On some expeditions, kites were used as towing devices on the glacier plateau. In good winds, it made the journey faster. The comparatively low travel speed, on the other hand, is not a measuring problem because the aim is to keep the distances between the measuring points as short as possible. Of course, this type of expedition requires the willingness of the participants to face the physical demands. But this has never been different throughout the history of polar research, from the expeditions of the pioneers to the present day.

## 2. Materials and Methods

The basic measurement in this project was the use of GNSS. The theory of GNSS is described in many technical articles, as well as the development of accuracy [23–25]. In high geographical latitudes, the integration of the GLONASS navigation system proves to be advantageous [26,27].

In our case, for the static GNSS measurements from 2002 to 2015, different generations of Trimble antennas and receivers were used. Since 2015, additional kinematic GNSS measurements have been performed using the NavXperience 3G + C antenna with the Trimble R7 receiver in 2015. Subsequent expeditions used the combined Trimble R10 and R12 systems. Portable GNSS units were used for orientation on all expeditions. Signals in the L- and G-band range of GPS and GLONASS (later also BeiDou and Galileo) were received. The accuracies of the campaigns are shown in Table 1. All measured coordinates are used with ellipsoidal heights.

During the nearly 40-day expedition, field logs were made of antenna heights as well as sled lowering depths and how they changed throughout the day. In addition to these geodetic records, weather data and density measurements were also noted (2017, 2020, 2021).

### 2.1. Static GNSS Measurements

Static GNSS measurements were taken approximately every 20 km at the respective overnight camps in 2002, 2006, 2010, 2012, and 2015, with each expedition member reaching the camp established in 2002 to ensure comparability. Upon arrival, the antenna was set up, aligned, and connected to the receiver (see Figure 3). The system is powered by solar cells. The measurement time is between 8 and 12 h, and the equipment is stored at a sufficient distance to avoid interference with the signal. The earlier measurements were made using ground stations on both coasts (Kangerlussuaq and Kulusuk and Tasiilaq, Kangerlussuaq and Ilulissat, respectively), and later, precision was achieved using Precise Point Positioning (PPP) by correcting the orbits afterward. WGS84 was used as a reference frame.

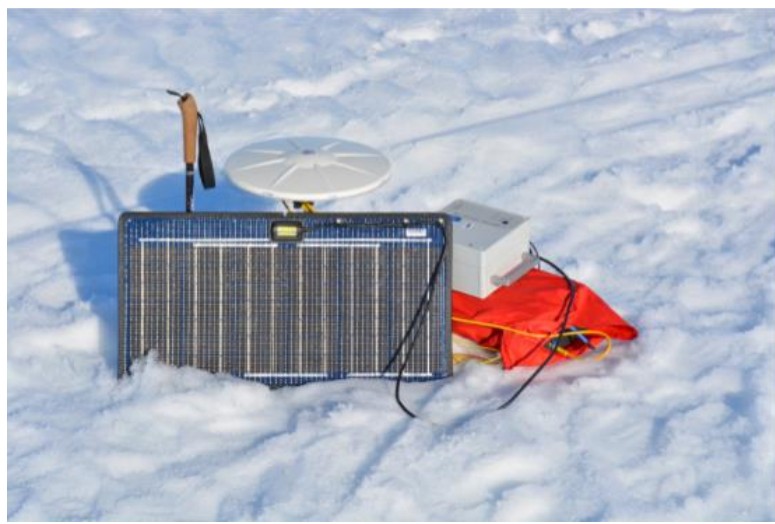

**Figure 3.** Static GNSS measurement during GKE 2015.

## 2.2. Kinematic GNSS Measurements

Kinematic GNSS measurements took place in 2015, 2017 (east coast only), 2020 (about 500 km), and 2021. The antenna was mounted on the pulka of an expedition member, and there were second-by-second recordings of the individual GNSS points. The 2 systems, GPS and GLONASS, were used (see Figure 4).

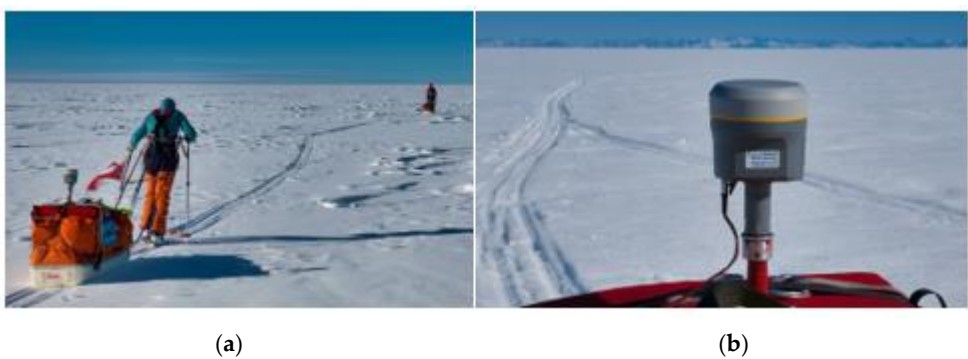

(**a**)           (**b**)

**Figure 4.** Kinematic GNSS measurement during GKE 2020 (**a**) on the pulka. (**b**) Trimble R12.

During post-processing, the TEQC software quality check was performed to verify the quality of the obtained data and to adjust the approximate position in the header of the observation files *.yyO of the RINEX data [26]. Because Greenland is a remote location and the technology has evolved, no extension systems or ground stations were used as reference stations for the kinematic measurements. Precise Point Positioning (PPP) in the International Terrestrial Reference Frame (ITRF2014) is used to achieve the precision of the data. Natural Resources Canada (NRCan) is used as the provider. Uploading is done through a web interface, and corrected positions are sent by mail. Using accurate ephemerides and clock corrections, the position can be determined to the nearest centimeter [28]. It takes 13 days to calculate the final corrections, so this period should be weighted between data collection and precision. Care should also be taken to use the correct evaluation method (static/kinematic) with the associated data, to always have enough satellites available, and to minimize the individual error sigma.

In the next step, the plate kinematics are considered, using 01.06.2015 (00:00:00) as the reference date, and the corresponding displacements and rotations of the North American plate are included in the plate motion model according to ITRF2014. This allows us to compensate for the effect of the Glacial Isostatic Adjustment (GIA) in Greenland [29].

After the data have been specified and reduced, they are further processed with Matlab. An overview of the program flow can be seen in Appendix A. First, the entries for the sled sinking depth and the antenna height are taken from the field book records. This is followed by a temporal sort and subsequent low-pass filtering of the data using Gaussian filtering to minimize the influence of noise. After a parameter study, a filter order of m = 50 is used as a target for the local topography of the Greenland ice sheet [30–32]. Due to the second-by-second measurement points and the largely tall jump-free relief, more distant points can also be considered for smoothing.

Two different principles are used to compare the GNSS kinematic data. First, crossing points are investigated, which requires a linear interpolation of two data points at the same position coordinates. Second, the principle of minimum distances between each data point of the 2015 expedition and subsequent expeditions is considered [32–36].

### 2.2.1. Crossing Point Comparison

For this purpose, the data are converted into the appropriate format so that they can be read and processed by the Linux-based program Generic Mapping Tool (GMT). The crossing points are determined as a linear interpolation between two different years. It should be noted that these are calculated values and not measured values. However, the advantage is that the position coordinates match exactly, and local unevenness has less influence. Since there are no large jumps on the Greenland ice sheet, the method is well suited. The x2sys package included in GMT is used for the calculation. The obtained crossing points are transformed to UTM coordinates in order not to neglect the curvature of the Earth. Then, the crossing points are assigned to the continuous track of the 2015 expedition by using the closest data point in each case. This method is sufficiently accurate over the entire track of nearly 700 km. In each case, the distances within a UTM zone are searched. The altitude differences previously calculated with GMT can now be visualized and analyzed.

### 2.2.2. Comparison of the Minimum Distances

For the comparison over the minimum distances, UTM coordinates are also used, and the holding times, which are caused, e.g., by pauses, are eliminated. As tolerance for the elimination of values, the distance of 0.005 m is used. Thus, the total matrix can be slimmed down considerably, and the computation time is shortened enormously. With this method, the distances of the position coordinates of an expedition to those of a following expedition are determined, and afterward, the respective data point with the smallest distance to the reference distance (here: 2015) is assigned. The calculation is very time-consuming and can be significantly shortened by using multiple processors via parallel computing in Matlab. After each point is assigned a minimum distance to a point in the follow-up measurement, data that are above tolerance are truncated. For the Greenland ice sheet, this was chosen for 5 m after the completion of the parameter study.

In addition, an adjustment to the data was made for the crossing point comparison. In the seasonally comparable expeditions in 2015 and 2020, almost 200 km were missing on the west coast because the expedition had to be aborted prematurely. However, in the following May 2021, the route could be walked completely, so the elevation component of the coordinates on the west coast is shifted to the connection point. The further one moves away from the endpoint of the 2020 expedition, the greater the uncertainty in the result becomes. Finally, the differences obtained are plotted and illustrated using Matlab's mapping toolbox.

## 3. Results

### 3.1. Static GNSS Measurements

Static measurements were made at 36 points spaced about 20 km apart. Figure 5 shows the mean annual elevation change at these points.

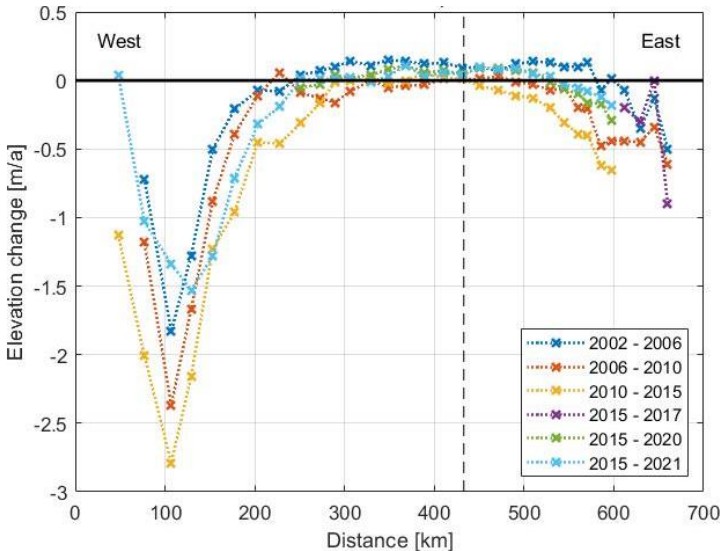

**Figure 5.** Annual elevation changes in the camps from 2002.

The catchment areas of the two glaciers Helheim (East coast in Figure 5) and Ilulissat Isbræ (West coast in Figure 5) are clearly visible. The watershed is located at km 420 (in Figure 5) and represents the highest point along the route. The 2002, 2006, 2010, and 2015 measurements seem to indicate an acceleration of mass loss, but the 2020/21 measurement does not confirm this. It appears that longer time series are needed to identify more reliable trends.

Figure 6 shows the absolute elevation changes at the camps between 2002 and 2021. While there is hardly any increase in the accumulation area (max. 2.1 m), there is an elevation loss of max. 38.7 m in the reservoirs in the Ilulissat Isbræ catchment. An elevation decreases of max. 6.4 m in the comparable area in the Helheim Glacier catchment and 10.1 m in the marginal area of the ice sheet on the east coast can be seen in Figure 6.

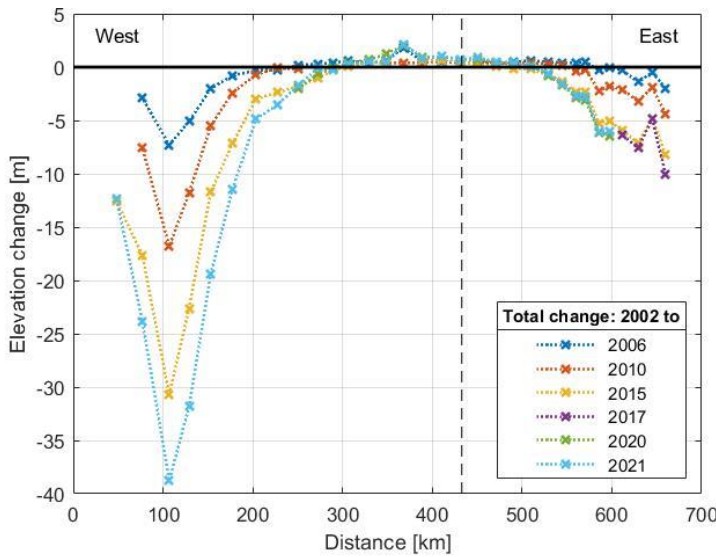

**Figure 6.** Total elevation changes in the camps from 2002.

### 3.2. Kinematic GNSS Measurements

3.2.1. Profile Comparison

First, the elevation profiles are compared. These retain their rough but finer details, which can be seen when magnified (Figure 7). As expected, the ice elevation decrease is

more pronounced along the coasts, which is enhanced by the two outlet glaciers Helheim (fastest flowing outlet glacier on the east coast of Greenland at approx. 30 m/d and Jakobshavn Isbræ (the most productive glacier on the west coast) since the expedition route lies within the influence of these [28]. However, when local ice elevation topography is considered, slight terrain elevations show a larger elevation change than the adjacent depressions (Figure 7). Overall, individual values fluctuate up to +/−30 cm per year around a sectionally stable mean or median.

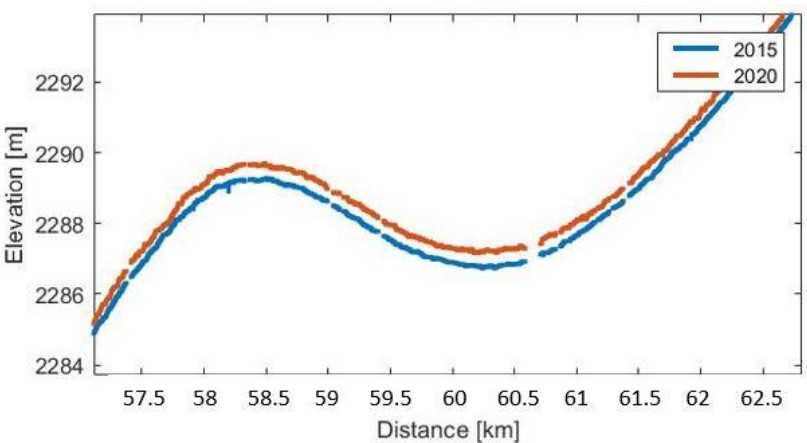

**Figure 7.** Comparison of a selected part of the profile from 2015 and 2020.

### 3.2.2. Elevation Change Comparison

With the help of the crossing point comparison, between 300 and 3000 intersections could be found, depending on the comparative section of the respective expedition, providing a dense network of data over the entire route. On the sections covered by skis and pulka, the density is significantly higher than in the sections covered by the kite. In practice, it is easier to generate crossing points in these areas because the speed traveled is lower. Particularly in the marginal area of the ice sheet, the variance of measured values is larger when comparing the expeditions' data, which is mainly due to the surface topography, as it is characterized by meltwater channels (see Figure 8) at the time of most expeditions (except 2021). In addition, the change from the minimum distance calculation is added here. After applying the previously described cutoff rule with a tolerance of 5 m, significantly more comparison points remain than for the crossing points. Occasionally, measurement gaps occurred during the expeditions, so no comparison is possible at these points.

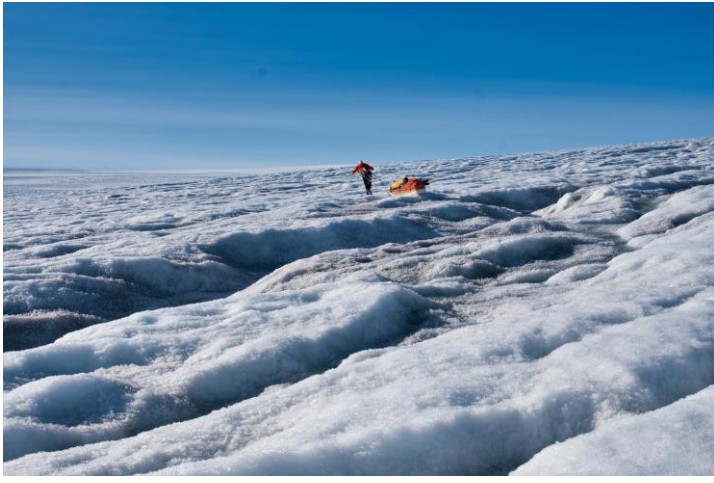

**Figure 8.** Meltwater channels during the expedition 2020 on the east coast.



The elevation change produces very similar results with both methods (Figure 9). Statistical values such as the mean and median deviate only by 2–3 cm in selected sections. However, the method is not generally valid in this form. Only because of the known topography of the ice sheet with few slopes does it remain very reliable. However, it can be assumed that the error in the method of minimum distances is larger than in the method of the crossing point comparison.

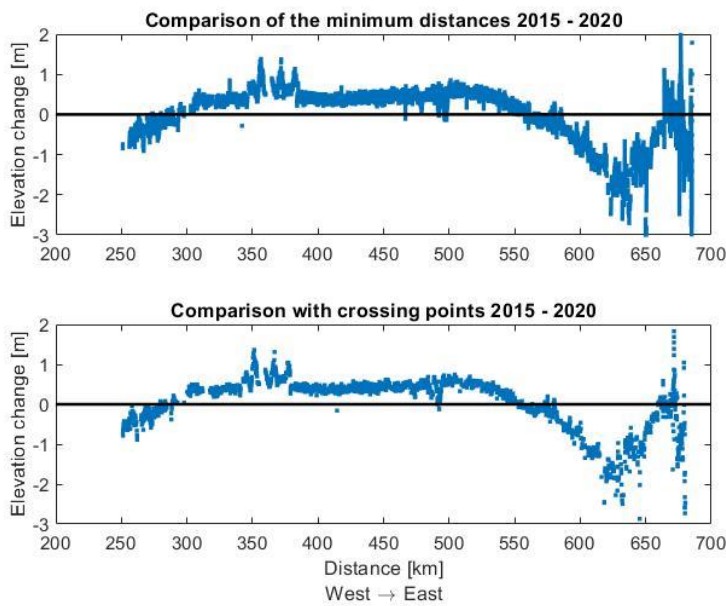

**Figure 9.** Comparison of the minimum distances method with 5 m tolerance (**top**) and the crossing point method (**bottom**).

### 3.2.3. Seasonal Changes during the Winter

The 2020 expedition took place in August/September and the next in the following May 2021 so that the seasonal changes could be observed over the winter. In Figure 10, these changes are shown along the profile, with the elevation component almost constant in the ice center. Towards the coasts, an increase due to precipitation of up to 2 m is observed. The west coast could not be investigated in more detail because the expedition had to be terminated prematurely.

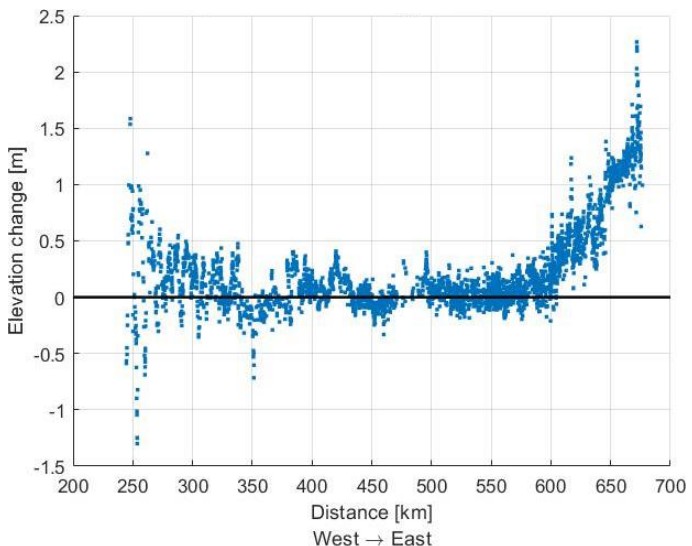

**Figure 10.** Elevation changes between before winter (August 2020–September 2020) and after winter (May 2021). Difference (blue points) = 2020–2021.

### 3.2.4. Modification of Missing Parts

For the kinematic data, complete profiles were only measured in 2015 and 2021, with the 2021 expedition taking place as early as May rather than between late July and early September as all previous expeditions had. An expedition took place in the previous season, covering about 500 km, so a link to the data from 2021 is made in this step to obtain a complete and seasonally comparable data set. Figure 11 shows the crossing point comparison for both 2015 and 2020 as well as 2015 and 2021, where the data density is not quite as high due to flooded sections.

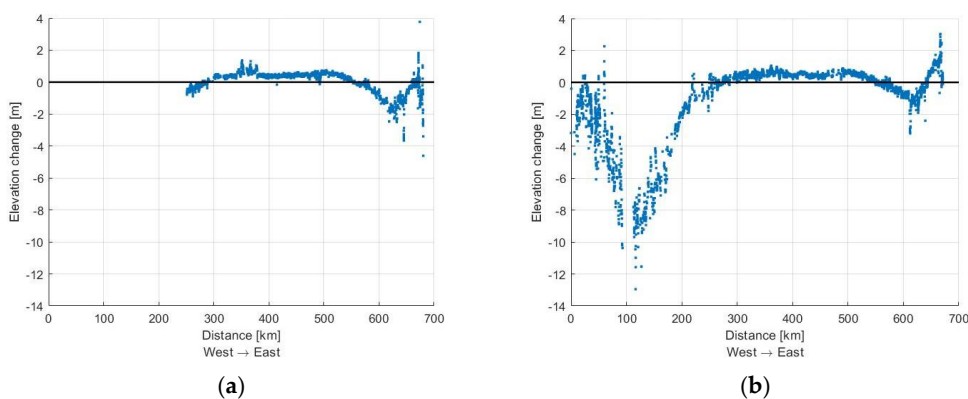

**Figure 11.** Comparison of crossing points (**a**) 2015–2020 and (**b**) 2015–2021.

The change in elevation over the winter from 2020 to 2021 is accounted for by appending to the endpoint of the 2020 data and the 2021 data set and shifted by the difference of $-0.7448$ m to get more realistic results for the coastal area (Figure 11). The farther the data is from the connection point, the larger the inaccuracy becomes.

In Figure 12, as in the static measurements, it can be seen that the influence of the two heads of glaciers is clearly visible in the data set. A larger ice elevation decrease is expected near the coast, which is amplified by calving the glaciers. Near the east coast, the decrease is somewhat delayed, which is probably related to the damming effect of the Schweizerland Alps (mountains on the east coast of Greenland, as de Quervain called them).

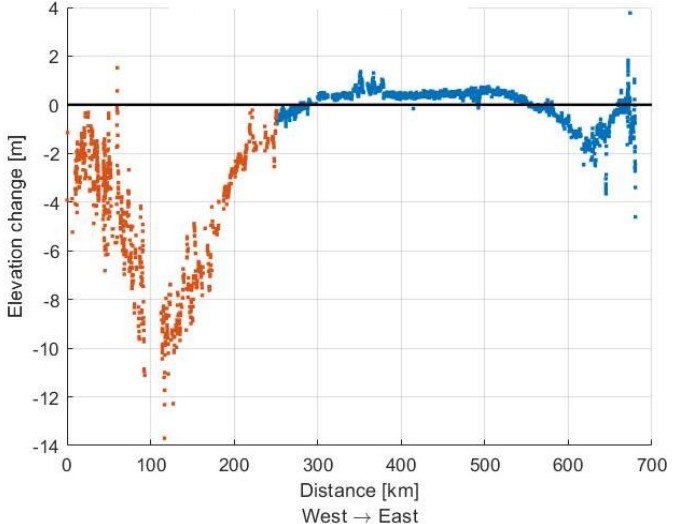

**Figure 12.** Comparison of crossing points along the expedition route with modified elevation change at the west coast. Blue color = 2020; orange color = 2021.

### 3.2.5. Accumulation and Ablation

The data show accumulation and ablation along the profile (see Figure 13). As expected, there is accumulation in the central part of the ice sheet and ablation toward the coasts. It should be noted that the seasonal change shifts the equilibrium line. It also illustrates the influence of the Helheim Glacier and the Jakobshavn Isbræ, with a catchment area of up to 200 km inland. For comparison, the ice velocity in Greenland was determined in [17] by Sentinel-1. The extensive catchment areas of the glaciers in relation to the expedition route are shown in Figure 14.

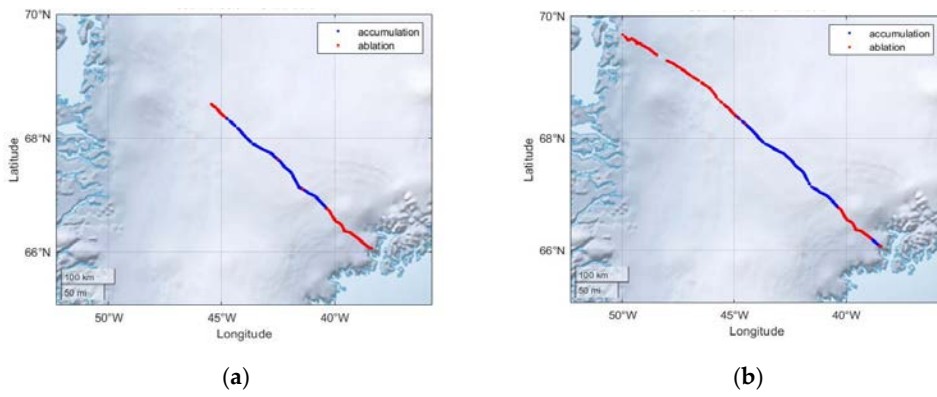

(**a**)　　　　　　　　　　(**b**)

**Figure 13.** Accumulation (blue) and ablation (red) along the expedition route based on data from (**a**) 2015–2020. (**b**) 2015–2021.

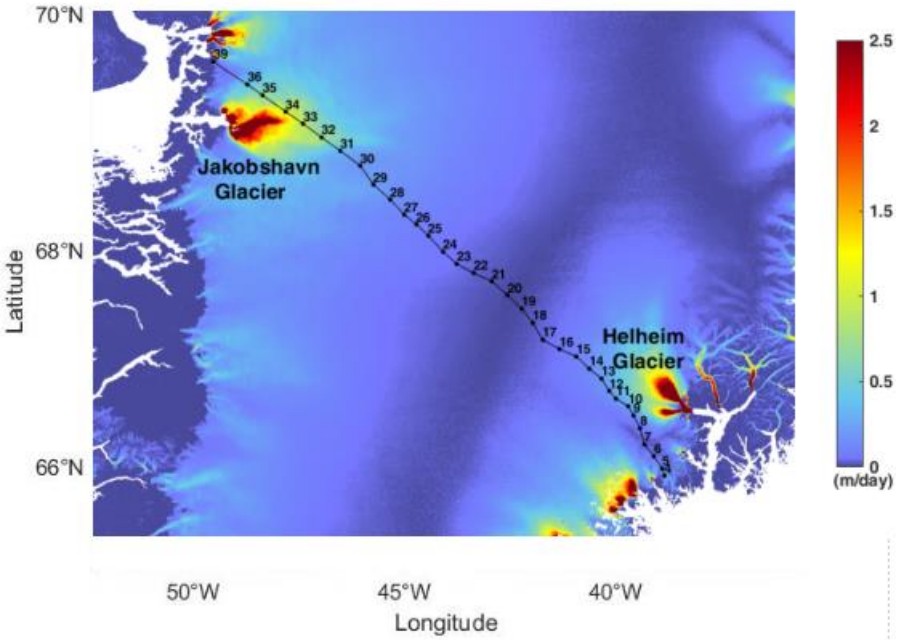

**Figure 14.** Ice velocity from synthetic aperture radar of Sentinel-1 acquired over October 2015–September 2016 with camps of the Greenland Korth Expedition route. The direction of glacier flow is from the central axis of the Greenland ice sheet towards the coast [17].

### 3.2.6. Comparison with Other Data

Satellite altimetry provides a real information on glacier elevation. However, these are only partially usable. There are gaps of several kilometers between the ground satellite tracks (see Figure 15). A direct comparison of our traverse data with those of the satellite altimetry is only possible at the crossing points. The Geoscience Laser Altimeter System

(GLAS) is developed for the IceSat mission and has a precision of about 3 cm for a footprint with less than 80 m diameter [37].

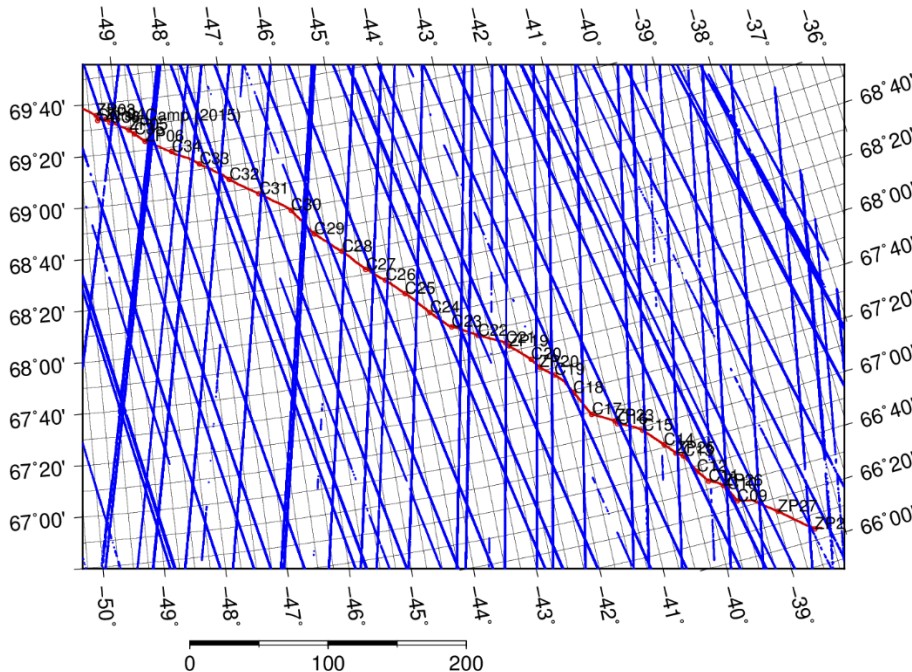

**Figure 15.** Coverage of the operation area by the IceSat mission (2003–2010). The diagonal line represents the profile measured by GKE on the ground.

The NASA IceSat satellite (Ice, Cloud, and land Elevation Satellite, 2003–2010) data were used. Comparing the profiles measured in this project with an elevation model derived from IceSat data [29], the qualitative difference between the two data sets becomes clearly visible (Figure 16). The elevation model is from 2010, based on the end of the IceSat mission and shows considerable deviations from the profile measurements due to the different spatial resolution. The IceSat provides a set of laser pulses, which have approximately 70 m spots on Earth's surface with a spacing of 170 m. The model from IceSat is interpolated and smoothed (blue line) compared to the profile measurement, which is more detailed (red line).

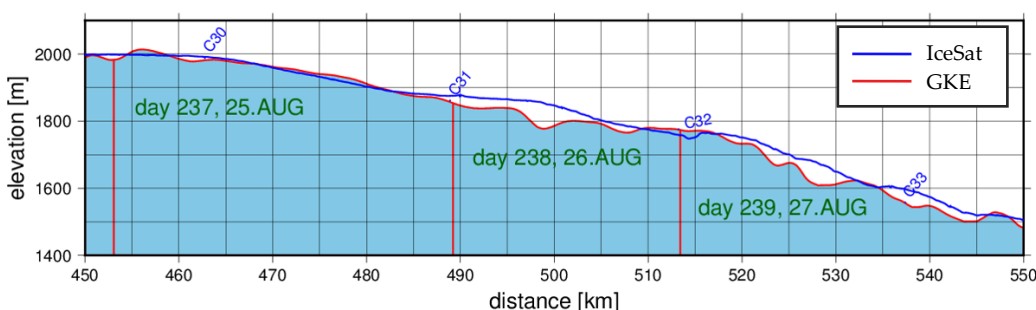

**Figure 16.** Part of the elevation profile from the west side (2015).

C30, C31, C32 and C33 are some of the points measured multiple times since 2002. The waves in the surface profile with amplitudes of up to 20 m are clearly visible. The blue line shows the heights derived from IceSat data; the red line shows the measured profile from this project.



## 4. Discussion–Error Influences

The aim was to demonstrate the possible link between terrestrial static or kinematic GNSS measurement and data derived from the GLAS device onboard the IceSat-1 satellite and, next, to verify the reliability and accuracy of the digital surface model based on IceSat-1 satellite measurements. The measured values were examined for random jumps using the difference quotient. There are jumps of a maximum of 1.2 cm, so that jump measured values cannot be identified as a significant source of error. Moreover, these are eliminated as much as possible by Gaussian filtering.

After PPP evaluation, the position accuracy is $+/-3$ cm, and the height accuracy is $+/-5$ cm for a single data set. For the investigations performed here, the elevation component is most relevant. Of course, there are some defined outliers; during measurement at these locations, a lower number of GNSS satellites occurred (PDOP). This influence is also visible in the dispersion of the elevation component (Figure 11, comparison km 680) around kilometer 680. This is classified as not trustworthy.

Regarding the measured antenna heights, a deviation of $+/-1$ cm is to be expected when it is attached to the pulkas. Likewise, variations of nearly 0 cm (in good conditions) to $+/-5$ cm (in uneven terrain) due to the ground conditions can be seen.

## 5. Conclusions

Geodetic-glaciological field work for monitoring glaciers and ice sheets is necessary even in the age of satellite technology. On the one hand, it is for the verification of the satellite data, but on the other hand, they also provide important results of their own.

The elevation changes determined during the Greenland Korth Expeditions (GKE) show a continuous melting process since 2002, mainly on the west coast. On the east side, the amount has increased from 20–40 cm/yr to 40–80 cm/yr. On the west side, the maximum annual ice loss has increased from 1.7 to 2.7 m/yr. Overall, the ice elevation at profile kilometer 100 has decreased by more than 35 m since 2002.

It turns out that long-term observations are always needed to make claims about climate change. Our observations are based on a historically short period of time, about 100 years. Nevertheless, it can be argued that we are now observing an enormous melting of Arctic ice, especially on the west coast of Greenland. It is not the purpose here to discuss causes or consequences, although this may have far-reaching implications for climate change, a possible change in the Gulf Stream, sea level rise and thus a significant impact on humanity. The aim was to demonstrate the possible link between terrestrial static or kinematic GNSS measurement and data derived from the GLAS (Geoscience Laser Altimeter System) device onboard the ICESat-1 satellite and, next, to verify the reliability and accuracy of the digital surface model based on ICESat-1 satellite measurements. A relatively good agreement was achieved, the differences being due to the different resolutions and the different terms of observation of the Greenland ice sheet.

**Author Contributions:** Conceptualization, T.H. and L.N.; methodology, T.H. and L.N.; software, L.N.; validation, T.H. and L.N.; investigation, T.H; data processing, L.N. and T.H.; writing—original draft preparation, T.H., L.N. and K.P.; writing—review and editing, T.H., L.N. and K.P.; visualization, T.H. and L.N.; supervision, K.P.; project administration, T.H. and K.P. All authors have read and agreed to the published version of the manuscript.

**Funding:** This research was funded by the Brandenburg University of Technology Cottbus-Senftenberg, Faculty of Architecture, Civil Engineering and Urban Planning. From the Czech part, the project was supported by a grant SGS22/049/OHK1/1T/11.

**Institutional Review Board Statement:** Not applicable.

**Informed Consent Statement:** Not applicable.

**Data Availability Statement:** The used data were collected during the expeditions and from open sources.

**Acknowledgments:** Our special thanks go to Wilfried Korth, who initiated the scientific project and actively accompanied it for many years. Unfortunately, he died in an accident in 2019. He was

always involved in the expeditions carried out until then and had the research and data in his hands. Furthermore, we thank Carsten Grienitz from AllTerra Germany GmbH for the support with Trimble technology. Finally, we thank Frank Polte and Marco Schütze, our expedition members in 2020. From the Czech part, the project was supported by a grant SGS22/049/OHK1/1T/11.

**Conflicts of Interest:** The authors declare no conflict of interest. The funders had no role in the design of the study; in the collection, analyses, or interpretation of data; in the writing of the manuscript, or in the decision to publish the results.

## Abbreviations

The following abbreviations are used in this manuscript:

| | |
|---|---|
| GIA | Glacial Isostatic Adjustment |
| GKE | Greenland Korth Expedition |
| GMT | Generic Mapping Tool |
| ITRF | International Terrestrial Reference Frame |
| NRCan | Natural Resources Canada |
| PPP | Precise Point Positioning |
| RINEX | Receiver Independent Exchange Format |
| UTM | Universal Transverse Mercator |

## Appendix A

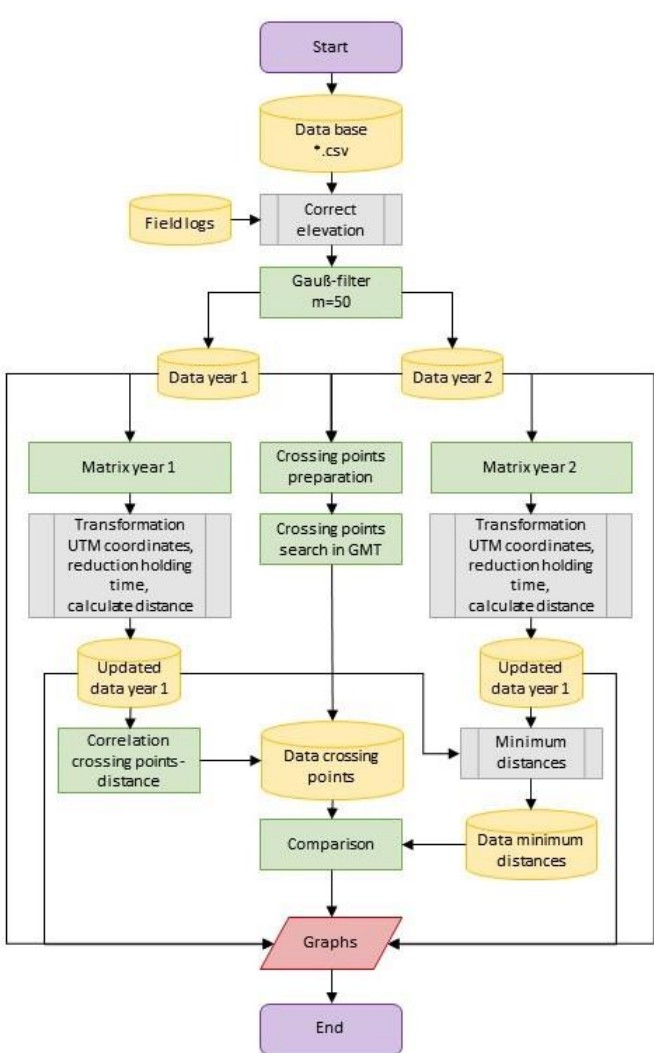

**Figure A1.** Program flow for kinematic analysis.

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
