# Peer review of "Ice Elevation Change Based on GNSS Measurements along the Korth-Traverse in Southern Greenland"

_applsci, doi:10.3390/app122312066_

Round 1

Reviewer 1 Report

Title - "GPS" is useless in the title

In general - improve the figures quality

Fig. 2 – is it possible to draw the border between the glaciers Helheim and Ilulissat to figure?

row 196 – differ mentioned glaciers by vertical line in figure 5. “Left and right” is not clean.

Fig. 5 – last two period should be 2017-2020 and 2020-2021?

In chapter 3.1 – Static GNSS measurement - you use the kinematic measurement too. Is it ok? Or change title of chapter.

Fig. 7 – decrease vertical range (1525 -1540 m is enough) and replace dots with line in legend

Fig. 9 – cutoff first 200 km. For comparison may be both methods can be in same graph. But I don’t know how it will be clear. May be, it could be useful vertical borders cutoff on 2 m. So the title of figure could be better: e.g.: Comparison of Minimum distances method (top) and Crossing point (bottom) with 5m tolerance.

Fig. 10 - cutoff first 200 km and spread graph in X direction. Is the 4m observation (top of the graph) ok or outlier? I recommend cutoff the graph in vertical direction too.

Are some outliers in all observations in general?

Row 253 – what is the value of difference, why shifted?

Fig. 14 – ice velocity, what is direction of movement (probably to the coasts?)? You can mention it in paper or draw to figure.

Row 278 – what is precision of IceSat elevation measurement? Please add it.

Fig. 16 – height/ better elevation, add the legend to the figure (red and blue line). I think “Part” is better than “Section” in the title of figure. What is the red line? Why there are so great differences? From which time period is IceSat elevation model presented?

Row 302 – “from 20 - 40cm/yr to 40 -80cm/yr” – from 20 cm/yr to 80 cm/yr

Row 311 – ground and satellite data – what do you mean ground - GNSS? I thing they are still satellite data. What do you mean satellite, Ice Sat? They are great differences. Please correct/explain it in conclusion.

Author Response

Thank you for your review, please, see comments.

Reviewer 2 Report

Review of the paper “Ice elevation change based on GPS/GNSS measurements along the Korth-Traverse in Southern Greenland”

General thoughts

In this paper, the authors present elevation changes determined during the Greenland Korth Expeditions (GKE). Results reveal a continuous melting process since 2002 mainly on the west coast. On the east side, the amount has increased from 20 - 40cm/yr to 40 -80cm/yr. On the west side, the maximum annual ice loss has increased from 170 to 270 cm/yr. Overall, the ice elevation at profile kilometre 100 has decreased by more than 3,500 cm since 2002. It turns out that long-term observations are always needed to make claims about climate change.

Detailed comments

Text is written in a logical and thoughtful way, creating a coherent whole. In general, it is written in accordance with the writing regime of a scientific paper (IMRaD). However, a few remarks must be discussed:

1)    Page one: cut "affiliation" in:

Affiliation 1: Chair of Mechanics and Numerical Methods, Institute of Civil and Structural Engineering, Brandenburg University of Technology Cottbus-Senftenberg, Germany

2 Affiliation 2: Department of Civil and Mechanical Engineering, Technical University of Denmark, Denmark

3 Affiliation 3: Department of Geomatics, Faculty of Civil Engineering, Czech Technical University in Prague, Czech Republic

2)    The authors should consider changing the abstract. The introduction is about research conducted by other scientists - it is not clear which sentences are about their own research and what is the subject of the research and what conclusions they have.

3)    The “GPS” in the title is not necessary.

4)    Line 15: It is accepted that farewell notes/posthumous note//etc. are placed at the end of the text, not at the beginning. I suggest moving to the acknowledgements section.

5)    Line 21 – climate, not the world. The paper is not about changing the world, as a whole structure (geographical, sociological, political etc).

6)    What was your motivation for research on the Korth-Traverse in Southern Greenland? Could you please add it to the text?

7)    Line 127:  Should not be “*.xxO”?

8)    The core of the paper is the GNSS measurements. Unfortunately, this issue was presented very briefly, without good practice. There is no information about the accuracies of GNSS campaigns, especially the oldest ones, where the quality of precise products, e.g. orbits was smaller than now. No information about ionosphere issues and the “last centimetres” – tropospheric model. Probably the authors used automat computation by the Canadian GNSS facility with the whole benefit of inventory, but both iono- and tropo- issues are important in height determination. All the heights are ellipsoid, or have you transformed them into normal or orthometric ones? What is a reference frame exactly (name the model)? Have you transformed all the data into one frame and epoch? Why almost all items in the literature, related to GNSS measurements, were made by the authors? Is there any other source dealing with the GNSS observation on the high latitude? It would tell what is the accuracy of the authors' observations, and explain why such methods were adopted (mainly – the observation’s duration). It might be worth adding another reference about the usage of GNSS in similar, earth sciences issues, e.g.:

a)    Yu K et al. Spaceborne GNSS Reflectometry. Remote Sensing 2022; 14(7): 1605.

b)    Guerova G et al. GNSS Storm Nowcasting Demonstrator for Bulgaria. Remote Sensing 2022; 14(15): 3746.

c)     Chwedczuk K, et al. Challenges related to the determination of altitudes of mountain peaks presented on cartographic sources. Geodetski Vestnik 2022; 66(1): 49–59.

d)    Zheng, Y. et al, Analyses of GLONASS and GPS+GLONASS Precise Positioning Performance in Different Latitude Regions. Remote Sensing 2022, 14, 4640. https://doi.org/10.3390/rs14184640

e)    Godah Walyeldeen et al., Comparison Of Vertical Deformation Of The Earth’s Surface Obtained Using GRACE-Based GGMS And GNNS Data - A Case Study Of South-Eastern Poland. Acta Geodynamica et Geomaterialia, vol. 17, no. 2 (198), Prague 2020. https://doi.org/10.13168/AGG.2020.0012

The(d) paper is interesting. Shows that at high latitude solutions benefit from the GLONASS navigation system (middle latitude e.g. Europe, might be otherwise). Information about the GRACE mission and its evaluation by GNSS observations might be found in many works, this is an example.

A fine project (financed by private capital !), related to the topic may be found here:

https://www.cs.york.ac.uk/cvpr/project/leapp/

and their work should be considered also:

f)      Cooper, M. A et al, Unravelling the long-term, locally heterogenous response of Greenland glaciers observed in archival photography. The Cryosphere, 16, 2449–2470, https://doi.org/10.5194/tc-16-2449-2022

g)    Lewinska P. et al, Evaluation of structure-from-motion for analysis of small-scale glacier dynamics. Measurement, pp. 1-15. https://doi.org/10.1016/j.measurement.2020.108327

Try to look with a wider perspective and improve the description of GNSS methodology. The University of York project is worth citing, their work is impressive.

9)    Item no. 8 – what is it? Why there isn't any description?

10) The manuscript file (PDF) weighs 25.5 MB. It looks weird. Nowadays, downloading and opening it on any electronic device is not a problem, but there are reasons why it is not practised. A habit and convenience, above all. The final PDF should not exceed over 10 MB.

Author Response

(The authors gave the same response as above.)

Round 2

Reviewer 2 Report

Dear authors

I haven't noticed the answer for review, but changes in the text were performed. Unusual situation. Nevertheless, thank you for considering my remarks.

- line 147. Looks a little bite out of context. Maybe rewrite it to sth like "... which is why since 2015 the GNSS observation has been used." Adding GLONASS improves the measurement, so you add this system to the position determining. The sequence of cause and effect in the text will be preserved, even if you added GLONASS because you had such equipment at your disposal.

Figure 3 and 8 - wired frame (with dots) for this figure.